# Molecular Characteristics and Polymorphisms of Buffalo (*Bubalus bubalis*) *ABCG2* Gene and Its Role in Milk Fat Synthesis

**DOI:** 10.3390/ani13193156

**Published:** 2023-10-09

**Authors:** Fangting Zhou, Xinyang Fan, Xiaoqi Xu, Zhuoran Li, Lihua Qiu, Yongwang Miao

**Affiliations:** 1Faculty of Animal Science and Technology, Yunnan Agricultural University, Kunming 650201, China; fangtingzhou666@126.com (F.Z.); xinyangfan1@126.com (X.F.); 18746763963@163.com (X.X.); lizhuoran85@126.com (Z.L.); lihuaqiu666@126.com (L.Q.); 2College of Chemistry, Biology and Environment, Yuxi Normal University, Yuxi 653100, China

**Keywords:** buffalo, *ABCG2*, polymorphism, milk fat synthesis, overexpression, knockdown

## Abstract

**Simple Summary:**

This study focuses on the *ABCG2* gene, which is known to play a crucial role in secreting vitamins into milk and transporting xenotoxic and cytostatic drugs across the plasma membrane in cattle, mice, and humans. However, the specific role of this gene in buffaloes, especially its effect on milk fat synthesis in buffalo mammary epithelial cells (BuMECs), remains inadequately understood. In this study, we isolated and identified the full-length coding region of the buffalo *ABCG2* gene from the mammary gland in buffalo and analyzed its physicochemical characteristics, gene structure, conserved domains and motifs, and polymorphisms. This study found that the *ABCG2* gene is highly expressed in buffalo mammary glands and plays an important role in milk fat synthesis in BuMECs. These findings contribute to our understanding of milk fat synthesis and could have important implications for the dairy industry. This could benefit both farmers and consumers by helping to provide high-quality milk products.

**Abstract:**

The ATP-binding cassette subfamily G member 2 (*ABCG2*) serves crucial roles in secreting riboflavin and biotin vitamins into the milk of cattle, mice, and humans, as well as in the transportation of xenotoxic and cytostatic drugs across the plasma membrane. However, the specific role of the *ABCG2* gene in water buffaloes (*Bubalus bubalis*), especially its effect on milk fat synthesis in buffalo mammary epithelial cells (BuMECs), remains inadequately understood. In this study, the full-length CDS of the buffalo *ABCG2* gene was isolated and identified from the mammary gland in buffaloes. A bioinformatics analysis showed a high degree of similarity in the transcriptional region, motifs, and conservative domains of the buffalo *ABCG2* with those observed in other Bovidae species. The functional role of buffalo *ABCG2* was associated with the transportation of solutes across lipid bilayers within cell membranes. Among the 11 buffalo tissues detected, the expression levels of *ABCG2* were the highest in the liver and brain, followed by the mammary gland, adipose tissue, heart, and kidney. Notably, its expression in the mammary gland was significantly higher during peak lactation than during non-lactation. The *ABCG2* gene was identified with five SNPs in river buffaloes, while it was monomorphic in swamp buffaloes. Functional experiments revealed that *ABCG2* increased the triglyceride (TAG) content by affecting the expression of liposynthesis-related genes in BuMECs. The results of this study underscore the pivotal role of the *ABCG2* gene in influencing the milk fat synthesis in BuMECs.

## 1. Introduction

The ATP-binding cassette subfamily G member 2 (*ABCG2*), also known as breast cancer resistance protein, belongs to the ATP-binding cassette protein superfamily [1]. Initially discovered in multidrug-resistant human breast cancer cell lines, *ABCG2* confers resistance to chemotherapeutic agents by actively extruding compounds such as mitoxantrone, topotecan, and methotrexate from the cell [2]. The human *ABCG2* gene, located on chromosome 4, consists of 16 exons and 15 introns, encoding a protein of 658 amino acid residues [3]. Functioning as a transporter located on the cell membrane, *ABCG2* uses ATP hydrolysis for the active transport of extracellular material into the cell [4].

In bovines, the *ABCG2* gene is located on chromosome 6, with a coding sequence of 1977 bp that encodes 658 amino acid residues [4]. It plays a pivotal role in the secretion of riboflavin (vitamin B2) and other nutrients in milk [1,5]. The upregulation of *ABCG2* during lactation in the mammary glands of dairy cows implies its participation in active drug secretion into milk [6] and potentially in the synthesis and secretion of milk [7]. Notably, *ABCG2* significantly impacts the milk yield, milk protein percentage, and milk fat percentage in bovines [8,9,10]. A genome-wide analysis revealed SNPs within the *ABCG2* gene region affecting milk production traits in cows [11]. Furthermore, blocking *ABCG2* inhibits the proliferation of bovine mammary epithelial cells, suggesting its role in mammary epithelial cell proliferation [12].

Water buffaloes (*Bubalus bubalis*) have been domesticated for 3000–6000 years, holding significant economic importance in tropical and subtropical regions due to their contributions to dairy, meat, and draught purposes [13]. It is estimated that there are more than 200 million buffaloes in the world. Domestic buffaloes are classified into two categories: river and swamp buffaloes. The former are primarily utilized for milk production, with each lactating buffalo producing more than 2000 kg of milk per year, while the latter are predominantly employed for draught purposes, with each lactating buffalo producing 500–600 kg of milk per year [14]. Although buffaloes are critical for agricultural development, compared with other domestic animals, genomic evaluation studies in buffaloes are still in the developing stage [15]. Therefore, it is very necessary to study the functional genes of buffaloes. Peroxisome proliferator-activated receptor gamma (PPARG) is considered to be a central regulator in the milk lipid synthesis in the mammary glands of cows, goats, and buffaloes [7,16,17]. The *ABCG2* gene has been identified as a direct target gene of PPARG [18]. Whether *ABCG2* is involved in the synthesis and secretion of milk lipids in lactating buffalo mammary epithelial cells requires further investigation. This study aimed to investigate the buffalo *ABCG2* gene by isolating and identifying its complete coding sequence (CDS) and analyzing its gene structure, physicochemical properties, motifs, and functional domains through bioinformatics methods. Additionally, the tissue-specific expression of *ABCG2* was assessed using real-time quantitative PCR (RT-qPCR). Furthermore, polymorphisms in the coding region of *ABCG2* were detected and analyzed using the direct sequencing of PCR products and population genetic methods. To further understand its role in buffalo mammary gland lactation, *ABCG2* was investigated through lentivirus-mediated overexpression and knockdown in buffalo mammary epithelial cells (BuMECs). Ultimately, this study will provide insights into the molecular characteristics of buffalo *ABCG2* and its significance in mammary gland lactation.

## 2. Materials and Methods

### 2.1. Sample Collection

Five lactating Binglangjiang buffaloes (peak lactation, five years old, about 60 d postpartum) were slaughtered, and tissue samples of the heart, liver, kidney (medulla), lung, mammary gland, adipose tissue, muscle, rumen, small intestine, spleen, and brain (cortex) were promptly collected and flash-frozen in liquid nitrogen. In addition, mammary gland tissue samples were surgically obtained from five Binglangjiang buffaloes at sexual maturity (two years old) and five Binglangjiang buffaloes (five years old) that underwent four different physiological states: early lactation (approximately 20 days postpartum), peak lactation (approximately 60 days postpartum), late lactation (approximately 220 days postpartum), and dry-off period (approximately 60 days before parturition) following previously described methods [19]. RNA extraction was performed utilizing RNAiso Plus (TaKaRa, Dalian, China), and the cDNA synthesis was carried out utilizing the PrimeScript RT kit (TaKaRa, Dalian, China) with 2 µg RNA for each sample.

Furthermore, blood samples were collected from 102 healthy adult buffaloes, comprising 52 Binglangjiang buffaloes (river type) and 50 Dehong buffaloes (swamp type), for the purpose of SNP detection of the *ABCG2* gene. The buffaloes used for sample collection were all healthy adult buffaloes and were not related.

### 2.2. Isolation, Identification, and Bioinformatics Analysis of Buffalo ABCG2 Gene

To amplify the CDS of buffalo *ABCG2*, a pair of primers was designed based on the mRNA sequence of cattle *ABCG2* (accession no. XM_006042277): forward primer, 5′-CCAGCGAGATACTGTAGTT-3′; reverse primer, 5′-TCACTGAAATTAAAGAGGAA-3′. PCR used cDNA from mammary gland tissue as the template, following the manufacturer’s instructions for 2× PCR Master Mix (CWBIO, Beijing, China). After electrophoresis on a 1.5% agarose gel, PCR products were purified using the Gel Extraction Kit (OMEGA, Norcross, St. Petoskey, MI, USA). The purified PCR product was then cloned into the pMD-18T vector and sequenced bidirectionally by Shanghai Biological Engineering Technology Services Co., Ltd. (Shanghai, China). The obtained raw data were processed and analyzed using SeqMan and EditSeq in Lasergene 7 software package (v7.1.0) (DNAStar Inc., Madison, WI, USA). The open reading frame (ORF) of obtained sequence was determined using ORF Finder (http://www.ncbi.nlm.nih.gov/orffinder/, accessed on 5 June 2023), and homologous sequences were retrieved using the BLAST program (https://blast.ncbi.nlm.nih.gov/Blast.cgi, accessed on 5 June 2023) from the NCBI database. The basic characteristics, hydropathy, signal peptide, subcellular localization, transmembrane regions, and secondary and tertiary structures were determined using ProtParam (http://web.expasy.org/protparam/, accessed on 7 May 2023), ProtScale (http://web.expasy.org/protscale/, accessed on 7 May 2023), SignalP-5.0 Server (https://services.healthtech.dtu.dk/service.php?SignalP-5.0, accessed on 12 May 2023), ProtComp 9.0 (http://linux1.softberry.com/berry.phtml, accessed on 12 May 2023), TMHMM 2.0 (https://services.healthtech.dtu.dk/service.php?TMHMM-2.0, accessed on 12 May 2023), SOPMA (https://npsa-prabi.ibcp.fr/cgi-bin/npsa_automat.pl?page=npsa%20_sopma.html, accessed on 12 May 2023) and SWISS-MODEL (http://swissmodel.expasy.org/, accessed on 7 May 2023), respectively. A phylogenetic tree was constructed using Mega 7 [20] via the maximum likelihood method (the cpREV model) based on the amino acid sequences. The genome annotation GTF files from various species (buffalo: GCF_019923935.1; cattle: GCF_002263795.1; zebu: GCF_000247795.1; yak: GCF_000298355.1; bison: GCF_000754665.1; goat: GCF_001704415.1; sheep: GCF_016772045.1) were downloaded from NCBI Datasets (https://www.ncbi.nlm.nih.gov/datasets/, accessed on 5 August 2023) to add genetic structure information to the *ABCG2* transcripts using the TBtools software (v1.108) [21], followed by visualization using the Gene Structure Display Server 2.0 (http://gsds.gao-lab.org/, accessed on 7 August 2023). Conserved motifs in *ABCG2* proteins were identified through the MEME 5.5 website (http://meme-suite.org/tools/meme, accessed on 7 August 2023), and conservative domains were determined using NCBI Batch Web CD-Serach Tool (https://www.ncbi.nlm.nih.gov/Structure/bwrpsb/bwrpsb.cgi, accessed on 5 August 2023).

### 2.3. Tissue Differential Expression Analysis

The RT-qPCR primers used in this analysis are listed in Appendix A. For mRNA expression analysis, the geometric mean of the Ct values of *β*-actin (*ACTB)*, glyceraldehyde-3-phosphate dehydrogenase (*GAPDH*), and ribosomal protein S23 (*RPS23*) served as the endogenous control. RT-qPCR was performed on an Applied Biosystems™ 7500 (Thermo Fisher Scientific, Waltham, MA, USA) with TB Green^®^ Advantage^®^ qPCR Premix (Takara, Dalian, China). The purity of PCR product was confirmed through melting curve analysis, and the amplification efficiency was determined using LinRegPCR (www.linregpcr.nl, accessed on 7 May 2023; Appendix A). Relative expression levels of the gene in different tissues were evaluated using comparative method of 2^−ΔΔCt^ [22].

### 2.4. Genotyping of ABCG2 Polymorphisms

Genomic DNA was extracted from the blood samples using TIANamp Genomic DNA Kit (TIANGEN, Beijing, China). Primers for amplifying the DNA containing the coding region of buffalo *ABCG2* (accession no. NC_059163.1) were designed and are listed in Appendix A. The PCR reactions used the 2× PCR Master Mix (CWBIO, Beijing, China) following the manufacturer’s instructions. The authenticity and polymorphisms of PCR products were confirmed via direct DNA sequencing. Population genetic data analysis was conducted using PopGen32 software (v1.32) [23]. Haplotypes of observed SNPs were inferred via PHASE (v2.0) [24].

### 2.5. Construction of pEGFP-C1-ABCG2 Overexpression Plasmid

The overexpression plasmid, pEGFP-C1-ABCG2, containing the buffalo *ABCG2* gene, was generated through PCR using the pEGFP-C1 vector (Clontech Laboratories, Inc., Palo Alto, CA, USA). Specific primers were designed for PCR amplification, incorporating Xho I and Kpn I restriction sites (forward: 5′-CTCGAGATGCTCAAAATGTCATCCAATAG-3′; reverse: 5′-GGTACCTTAAGAAAATTTTTTAAGGAATAAC-3′; the restriction sites are underlined). Once constructed, the recombinant plasmid underwent sequencing for verification and was purified using the EndoFree Maxi Plasmid Kit (QIAGEN, Valencia, CA, USA) for cell transfection experiments.

### 2.6. Design and Cloning of ABCG2-Targeting shRNAs

Utilizing the online software, BLOCK-iT RNAi Designer (http://rnaidesigner.invitrogen.com/rnaiexpress/, accessed on 7 May 2023), three short hairpin RNAs (shRNAs) were designed to target *ABCG2* (Appendix A) based on the coding region sequences obtained in this study. These shRNA sequences were incorporated into the pLKO.1 vector to produce the final recombinant plasmids (pLKO.1-shRNAs). The recombinant vectors were sequenced for validation before being purified for use in cell transfection experiments using the EndoFree Maxi Plasmid Kit (QIAGEN, Valencia, CA, USA).

### 2.7. Cell Culture

Primary buffalo mammary epithelial cells (BuMECs), preserved in our laboratory, were cultured in DMEM/F12 medium (Gibco, New York, NY, USA). This medium was supplemented with 10% fetal bovine serum (Gibco, USA), 10 kU/L penicillin/streptomycin (Gibco, New York, NY, USA), 5 mg/L bovine insulin (Sigma, St. Louis, MO, USA), 5 mg/L hydrocortisone (Sigma, St. Louis, MO, USA), 1 mg/L epidermal growth factor (Sigma, St. Louis, MO, USA), and 5 μg/mL holotransferrin (Sigma, St. Louis, MO, USA). The cells were routinely cultured in a humidified incubator at 37 °C, 5% CO_2_, and 95% air. The culture medium was refreshed every 24 h. To induce lactogenesis, BuMECs were incubated in the above medium supplemented with 2 μg/mL prolactin (Sigma, St. Louis, MO, USA) for 24 h prior to experiments. Moreover, HEK-293T cells for generating lentivirus particles were acquired from Kunming Institute of Zoology, Chinese Academy of Sciences. These cells were cultured in DMEM/F12 medium containing 10% fetal bovine serum and 1% penicillin/streptomycin (10 kU/L, Gibco, New York, NY, USA).

### 2.8. Overexpression and Knockdown of Buffalo ABCG2 Gene

When BuMECs reached 70–80% confluence in culture plates, pEGFP-C1-ABCG2 (3 μg) was introduced into BuMECs using Lipo6000™ transfection reagent (Beyotime Biotechnology, Shanghai, China). pEGFP-C1 was transfected as a negative control. After 48 h of transfection, BuMECs were harvested for RT-qPCR analysis and TAG assay. The primer information for RT-qPCR analysis can be found in Appendix A. HEK-293T cells were cultured in 10 cm plates until they reached 70–80% confluence. The pLKO.1-sh1, pLKO.1-sh2, and pLKO.1-sh3 vectors were co-transfected into HEK-293T cells along with pEGFP-C1-ABCG2 to identify the shRNA with the highest interference efficiency. After 48 h of transfection, HEK-293T cells were collected for RT-qPCR analysis. Subsequently, pLKO.1-shRNA, pMD2G, and psPAX2 plasmids were co-transfected into HEK-293T cells at a ratio of 5:3:2 using Lipo6000™ transfection reagent to produce lentivirus (Lv-pLKO.1-shRNA). Negative control (Lv-NC) was established by transfecting pLKO.1-TRC, pMD2G, and psPAX2. After 48 h of transfection, the lentivirus was harvested, centrifuged at 1250 rpm for 5 min, and filtered through a 0.45 μm filter. The lentivirus particles were stored at −80 °C for long-term storage.

Subsequently, when BuMECs reached 70–80% confluence in the culture plates, Lv-pLKO.1-shRNA was added to the culture medium along with polybrene (2 μg/mL; Sigma, St. Louis, MO, USA) to enhance lentivirus infection efficiency. The medium was replaced with fresh medium 24 h later. After 48 h of infection, BuMECs were harvested for RT-qPCR and TAG assay.

### 2.9. Cellular TAG Content Analysis

After overexpression or knockdown of *ABCG2* for 48 h, the BuMECs were rinsed twice with PBS. The intracellular TAG concentration was assayed using the TAG kit (GPO-POD; Applygen Technologies Inc., Beijing, China) according to the manufacturer’s instructions. Simultaneously, the intracellular total protein concentration was measured using the BCA protein assay kit (Thermo Fisher, Waltham, MA, USA). The TAG content was then normalized per milligram of protein.

### 2.10. Data Analysis

All experiments were conducted with three biological replicates, and the data are presented as means ± standard error of the means (means ± SEM). GraphPad Prism 5 software (GraphPad Software Inc., La Jolla, CA, USA) was utilized for statistical analysis and data visualization. The statistical significance of differences between two groups was evaluated using two-tailed Student’s t-test. For multiple comparisons, one-way ANOVA with Tukey’s test was employed, and *p*-values less than 0.05 were considered statistically significant.

## 3. Results

### 3.1. Cloning and Identification of Buffalo ABCG2

The isolated *ABCG2* gene sequence from the buffaloes contained a full-length coding sequence (CDS) of 1977 bp, encoding a peptide comprising 658 amino acid residues. The comparison with other Bovidae species, including *Bos mutus* (XM_005897792), *Bison bison* (XM_010860190), *Bos taurus* (BT030709), *Bos indicus* (XM_019962487), *Capra hircus* (XM_018049143), and *Ovis aries* (GQ141082), showed a high sequence consistency, ranging from 97.47% to 98.94%. The sequence was deposited in the NCBI database with the accession number OK137537.1.

To explore the transcriptional structure of buffalo *ABCG2*, we compared all known transcripts of the gene in buffaloes with those of other Bovidae species. Among the eight transcripts identified in buffaloes, the coding regions contained either 14 or 15 exons, indicating an alternative splicing of the *ABCG2* gene (Figure 1). Specifically, the XM_025289519.1 transcript lacked the first exon in its coding region compared to the other buffalo transcripts, whereas the coding region of the buffalo *ABCG2* gene identified in this study contained 15 exons, and no coding region containing 14 exons was found. It is speculated that transcripts containing the coding region of 14 exons are not expressed in mammary tissue. In cattle, four distinct exon composition patterns were observed in the coding region: 13 exons (XM_024993324.1), 14 exons (XM_024993323.1), 15 exons (XM_010806035.3 and XM_024993319.1), and 16 exons (XM_024993311.1). Notably, the transcript patterns containing 14 and 15 exons in buffaloes were also found in cattle, corresponding to transcripts XM_024993323.1 and XM_010806035.3, respectively. Furthermore, variations in the 5′ untranslated region (UTR) and intron lengths were observed among different transcripts of the *ABCG2* gene within the same species, with even greater differences observed across species.

### 3.2. Characteristics and Structures of the ABCG2 Protein

Buffalo *ABCG2* had a theoretical pI of 8.80 and a grand average of hydropathicity of 0.188. It was computed to be stable with an instability index (II) of 30.39 and an aliphatic index of 101.66. Buffalo *ABCG2* lacked the N-terminal signal peptide, indicating that it is a non-secreted protein. Further prediction indicated that buffalo *ABCG2* contains six transmembrane helices (AA397–419, AA431–453, AA481–502, AA509–531, AA536–558, and AA632–654) (Appendix A) and is a potential membrane-bound endoplasmic reticulum protein with a score of 7.0.

The secondary structure analysis of buffalo *ABCG2* revealed a composition of 44.22% α-helix (291 AA), 16.72% extended chain (110 AA), 5.17% β turn (34 AA), and 33.89% random coils (223 AA) (Appendix A). Furthermore, the tertiary structure of buffalo *ABCG2* was predicted using SWISS-MODEL online based on homologous modeling (Appendix A). The sequence identity of *ABCG2* between buffaloes and humans (template: 6hbu.1A) was 85%, with a coverage rate of 99%.

The phylogenetic relationship and comparison of the motifs and conserved domains of *ABCG2* among buffaloes and other Bovidae species are depicted in Figure 2. In the phylogenetic tree, buffaloes were clustered with yaks and bisons on one branch, and cattle and zebu on another, while goats and sheep form a separate group (Figure 2A). This indicates a closer genetic relationship between buffaloes and species of the *Bos* genus compared to goats and sheep. The results of the motif pattern showed that all transcripts had all 10 motifs except for the XM_025289519.1 transcript in buffaloes and the XM_024993317.1 and XM_024993319.1–XM_024993324.1 transcripts in cattle, which had motifs 1–9 (Figure 2B). The *ABCG2* proteins across all Bovidae species were found to contain a 3a01204 domain, which belongs to the 3a01204 superfamily (Figure 2C). This finding indicates a functionally similar and conserved *ABCG2* protein among Bovidae species.

### 3.3. Analysis of the Expression Profile of Buffalo ABCG2

The mRNA expression profiles of *ABCG2* were analyzed to gain insight into its role in various tissues of Binglangjiang buffaloes. The highest expression levels were observed in the liver and brain, followed by the mammary gland, adipose tissue, heart, and kidney. Relatively lower mRNA expression levels were found in the small intestine, spleen, lung, muscle, and rumen (Figure 3A). To explore the potential impact of physiological stages on *ABCG2* expression, we assayed the mRNA levels in the mammary gland during the sexual maturity, early lactation, peak lactation, late lactation, and dry-off periods. Remarkably, the expression of *ABCG2* was significantly elevated during peak lactation, whereas its lowest expression occurred during sexual maturity (Figure 3B).

### 3.4. Population Variation Analysis

In this study, five SNPs were identified in the buffalo *ABCG2* gene, in which c.393 C>T and c.471 T>C were located in exon 5, c.720 C>T was located in exon 7, c.861 G>A was located in exon 8, and c.1290 C>T was located in exon 11 (Table 1). The sequencing results are shown in Appendix A. Interestingly, these five SNPs were exclusively found in river buffaloes, with no SNPs observed in swamp buffaloes. The Hardy–Weinberg equilibrium test showed that c.720 C>T and c.1290 C>T were in dis-equilibrium (*p* < 0.05). Notably, all SNPs were synonymous substitutions and did not lead to any amino acid changes.

### 3.5. Sequence Differences in ABCG2

Based on the SNPs identified in the *ABCG2* gene, a total of seven haplotypes (Buffalo_hap1–Buffalo_hap7) were defined in buffaloes (Appendix A). In addition, no new haplotype sequences were discovered in the published buffalo *ABCG2* sequences. Among these haplotypes, Buffalo_hap1 was shared by two types of buffaloes, and the rest were only found in river buffaloes. To delve into the sequence variations in the *ABCG2* gene, a comparison was conducted between the buffalo haplotype sequences and the published homologous sequences from other Bovidae species. Ten nucleotide differences were identified at the following positions: c.15, c.55, c.90, c.1017, c.1086, c.1278, c.1317, c.1542, c.1554, and c.1696 (Appendix A). Among these, four nucleotide differences (c.55, c.1086, c.1554, and c.1696) led to amino acid changes in the buffalo *ABCG2*, resulting in the corresponding amino acids: p.19Thr, p.362Asn, p.518Ile, and p.566Ser (Figure 4).

### 3.6. Overexpression of Buffalo ABCG2 Promotes Milk Fat Synthesis

Following the transfection of the pEGFP-C1-ABCG2 vector into BuMECs for 48 h, the mRNA expression of *ABCG2* exhibited a significant increase (*p* < 0.01) (Figure 5A). Accompanying the overexpression of *ABCG2*, several other genes related to lipid metabolism also demonstrated altered expression levels. Specifically, the gene expressions of *SCAP*, *PPARG*, *PPAPGC1A*, *SREBF2*, *INSIG2*, *FASN*, *ACC*, and *AGPAT6* significantly increased by 1.9-fold, 1.7-fold, 1.3-fold, 1.4-fold, 2.2-fold, 2.3-fold, 2.0-fold, and 5.7-fold, respectively (*p* < 0.05 or *p* < 0.01). Conversely, the expressions of *SREBF1* and *INSIG1* decreased dramatically by 28% and 30%, respectively (*p* < 0.05) (Figure 5C). Furthermore, the impact of *ABCG2* overexpression on the total triglyceride (TAG) content in BuMECs was investigated. The results demonstrated a significant increase in the TAG content (*p* < 0.05) upon *ABCG2* overexpression in BuMECs (Figure 5B).

### 3.7. Knockdown of Buffalo ABCG2 Inhibits Milk Fat Synthesis

To identify the shRNA with the highest interference efficiency against *ABCG2*, HEK-293T cells were co-transfected with pEGFP-C1-ABCG2 along with the pLKO.1-sh1, pLKO.1-sh2, and pLKO.1-sh3 vectors. The results demonstrated significant reductions in *ABCG2* expression compared to the control (pEGFP-C1). Specifically, the expression of *ABCG2* decreased by 44% (*p* < 0.01) with pLKO.1-sh1, by 76% (*p* < 0.01) with pLKO.1-sh2, and by 94% (*p* < 0.01) with pLKO.1-sh3 (Figure 6). These findings indicate that pLKO.1-sh3 exhibited the highest interference efficiency against *ABCG2*. Consequently, pLKO.1-sh3 was selected for lentivirus packaging.

After the co-transfection of pLKO.1-sh3, pMD2G, and psPAX2 in HEK-293T cells, the resulting lentivirus (Lv-sh3) was packaged and collected. Subsequently, the Lv-sh3 was transfected into BuMECs to achieve *ABCG2* knockdown. The mRNA abundance of *ABCG2* in the Lv-sh3 group was significantly lower (59%) than that in the Lv-NC group (*p* < 0.01; Figure 7A). The silencing of *ABCG2* resulted in altered expression levels of related genes, including decreases in *SCAP*, *PPARG*, *PPAPGC1A*, *SREBF2*, *INSIG2*, *FASN*, *ACC*, and *AGPAT6* expressions by 52%, 40%, 32%, 48%, 42%, 32%, 44%, and 59%, respectively, compared to the Lv-NC group (*p* < 0.01 or *p* < 0.05). Moreover, the expressions of *SREBF1* and *INSIG1* were significantly increased by 1.46-fold (*p* < 0.05) and 2.04-fold (*p* < 0.01), respectively (Figure 7C). Furthermore, the interference of buffalo *ABCG2* led to a significant decrease (*p* < 0.05) in the content of TAG in BuMECs (Figure 7B).

## 4. Discussion

In this study, the complete CDS of the *ABCG2* gene was successfully cloned from the mammary tissues of Binglangjiang buffaloes, and the obtained sequence exhibited a high homology of more than 97.47% with other Bovidae species. In addition, the transcriptional region of the buffalo *ABCG2* gene shared remarkable similarity with its counterparts of other Bovidae species. A phylogenetic analysis based on the *ABCG2* amino acid sequences revealed that buffaloes clustered with other species of the *Bos* genus, and the motifs and conserved structural domains of their *ABCG2*, were highly consistent. These findings collectively suggest that the function of buffalo *ABCG2* is likely similar to that of other Bovidae species. A bioinformatics analysis displayed that buffalo *ABCG2* is a non-secretory protein with six transmembrane helices and engages in the biological process of solute transport across lipid bilayers with ATP enzyme activity and ATP-enzyme-coupled transporter activity.

In the mammary glands of dairy cows, sheep, and goats, the *ABCG2* protein is primarily expressed in alveolar epithelial cells and most ducts. The highest expression is observed in the small intestine and mammary gland, with a high level in the liver and moderate amounts in the lung, colon, and kidney [25]. The results of this study showed that *ABCG2* expression was the highest in the liver and cortex of the brain of lactating buffaloes, followed by the mammary gland, adipose tissue, heart, and medulla of the kidney, while showing very low levels in the small intestine, spleen, lung, muscle, and rumen, which is slightly different from the results of the previous study [25]. We hypothesize that there are two reasons for this situation; one is that there may be species differences in the expression of the *ABCG2* gene, and the other reason may be related to the different physiological periods of the tissues. Whether this is the case needs further study. Furthermore, the expression of this gene was significantly higher during peak lactation in the mammary gland compared to that in the non-lactating mammary gland, which is consistent with the findings in dairy cows, sheep, and goats [25]. This reveals that buffalo *ABCG2* may be involved in the lactation process of buffaloes. In addition, the *ABCG2* gene was identified with five SNPs in river buffaloes, while it was monomorphic in swamp buffaloes. Although all five SNPs are synonymous substitutions, a previous study has shown that synonymous substitutions may alter the translation efficiency [26]. Therefore, we hypothesized that the expression of the *ABCG2* gene in the mammary glands of the two types of buffaloes is distinct, which may be one of the reasons for the differences in the lactation performance between the two types of buffaloes. In this study, the dominant alleles of the five SNP_S_ found in riverine buffaloes all have a tendency to be close to homozygous fixation, and this homozygous fixation trend has long been realized in swamp buffaloes, indicating that the *ABCG2* gene sequence of the ancestors of riverine buffaloes and swamp buffaloes is consistent. However, they may have undergone different nucleotide substitutions and genetic drift processes, resulting in the differences in their current nucleotide sequences. From another perspective, since the five SNP_S_ found in riverine buffaloes are all synonymous, indicating that the *ABCG2* amino acid sequence of riverine buffaloes and swamp buffaloes is identical, this reveals that *ABCG2* is functionally conserved in both types of buffaloes. It also revealed that the five SNPs found in this study may not have practical application significance in the breeding selection scheme of buffaloes.

Many studies in different dairy cow populations have identified *ABCG2* as a potential quantitative trait locus (QTL) related to milk production, including the milk yield, milk fat, and protein content [5,8,9,27]. However, its precise role in milk fat synthesis and secretion remained inconclusive. Bionaz et al. [7] pointed out that *ABCG2* plays an important role in secreting “some” important milk components, and is possibly involved in cholesterol transport, but the whole lactation period is not affected by cholesterol [28]. The only confirmed role of *ABCG2* in milk component secretion is in riboflavin, which is an essential but limited nutrient for newborns. Therefore, a significantly upregulated expression of this gene was observed during lactation, suggesting that *ABCG2* may possess additional functions beyond riboflavin secretion. Sterol regulatory element binding proteins (SREBFs) are members of the basic helix-cyclic-leucine zipper transcription factor family, regulating lipid homeostasis by controlling the gene expressions associated with cholesterol, fatty acids, TAG, and phospholipid synthesis [29]. When the cholesterol levels drop below a certain level, the binding of SCAP to the insulin-induced gene (INSIG) protein breaks down, releasing SCAP. Then, SCAP binds to SREBF to make SREBF an active transcription factor, which, in turn, activates the expression of its target genes to promote cholesterol synthesis. On the contrary, when the concentration of cholesterol is high, the INSIG-SCAP complex fails to activate SREBFs, thus inhibiting cholesterol synthesis and gene expression [29]. In this study, the mRNA abundance of *SCAP*, *SREBF2*, and *INSIG2* genes related to cholesterol synthesis increased significantly after the overexpression of the buffalo *ABCG2* gene in BuMECs. It is speculated that *ABCG2* might enhance cholesterol synthesis by aiding in cholesterol extracellular transport. Follow-up experiments need to be further verified. Our previous study has shown that the *PPARG* gene is an important core regulatory gene of milk fat synthesis, and the knockdown of *PPARG* led to a marked decrease in *ABCG2* expression within BuMECs [16]. With regard to the fact that the overexpression or interference of *ABCG2* will lead to the increase or decrease in the expression of the *PPARG* gene, we speculate that the regulation of *PPARG* by *ABCG2* may be a positive feedback loop. In the mammary glands of dairy cows, the profiles of *PPARG*, *SREBF1*, and *SREBF2* show high correlations with *ABCG2* profiles. And the increase in the *PPARG* levels at the onset of lactation reflects an elevated physiological demand for PPARG in the activation of *ABCG2* [30]. PPARG, PPARGC1A, and INSIG1 cooperate to regulate the function/expression of SREBF1 in the milk fat synthesis of dairy cows [7], which is consistent with the results of this study in which the expression of these genes had the same trend after *ABCG2* overexpression or interference. Fatty acid synthase (FASN) and acetyl-CoA carboxylase α (ACC) are activated by acyl-CoA synthetase (ACSS) to synthesize fatty acids [31]. This study also suggests that *ABCG2* may increase the de novo synthesis of fatty acids by upregulating the expression of *FASN* and *ACC.* In addition, the expression modulation of the *AGPAT6* gene related to triglyceride synthesis by *ABCG2* indicates its potential in regulating the milk fat content. This is corroborated by the observed changes in the triglyceride content upon the overexpression/knockdown of *ABCG2*.

## 5. Conclusions

In this investigation, we successfully isolated and characterized buffalo *ABCG2*, shedding light on its functional attributes. The structural integrity of the transcriptional region, as well as the presence of motifs and conserved domains, underscore the similarity between buffalo *ABCG2* and its counterparts in other Bovidae species. Notably, buffalo *ABCG2* exhibited prominent expression levels in the liver and brain, with the mammary gland following suit. Of particular interest, our findings unveiled a dynamic pattern of *ABCG2* expression in the mammary gland, with the highest level observed during peak lactation. This temporal variation indicates a potential role of buffalo *ABCG2* in lactation processes. Furthermore, the alterations in the mRNA expression observed in the key genes associated with milk fat synthesis, coupled with changes in the cellular triglyceride content, underscore the capacity of buffalo *ABCG2* to orchestrate milk fat biosynthesis within BuMECs. This study significantly advances our understanding of buffalo *ABCG2*’s role in modulating milk fat synthesis. By unraveling these mechanisms, we lay the groundwork for comprehending the genetic underpinnings and regulatory intricacies governing the milk fat traits in buffaloes.

## Figures and Tables

**Figure 1 animals-13-03156-f001:**
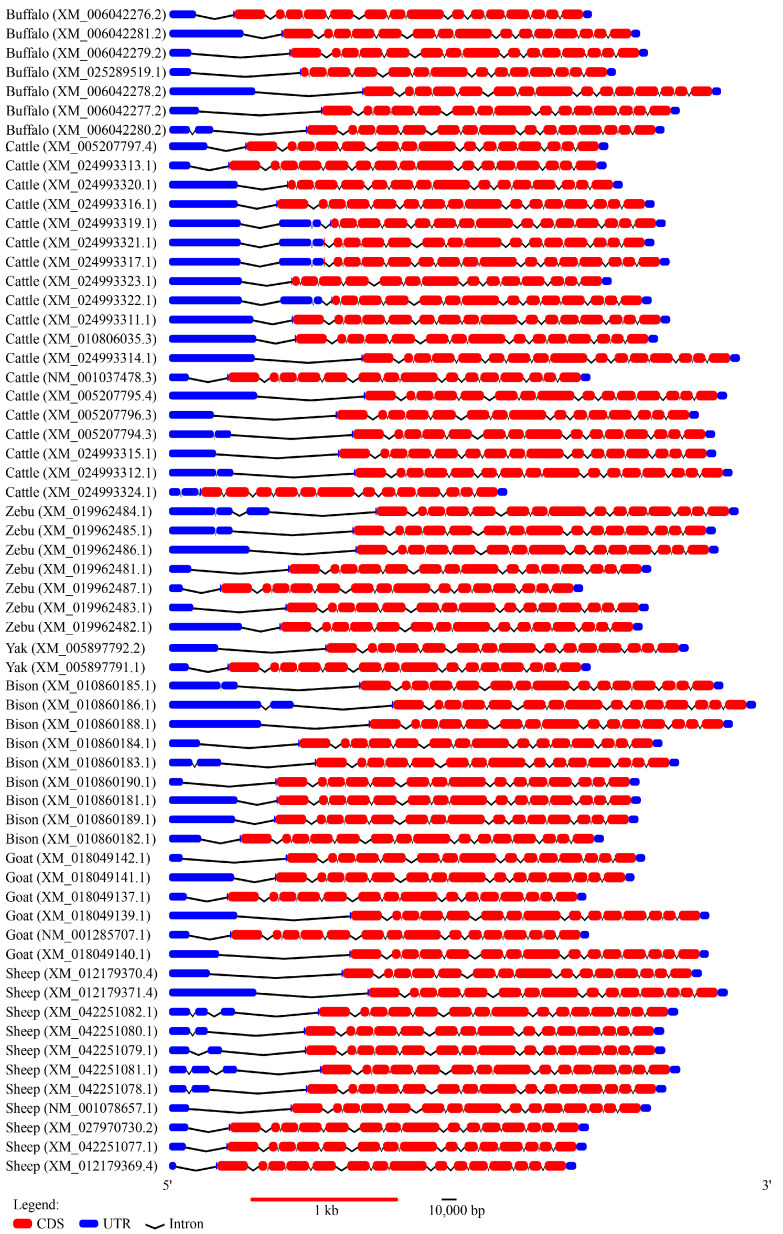
Transcriptional region structure of *ABCG2* in some Bovidae species. The coding region of the buffalo *ABCG2* gene obtained in this study belongs to the transcript type with 15 exons in the coding region.

**Figure 2 animals-13-03156-f002:**
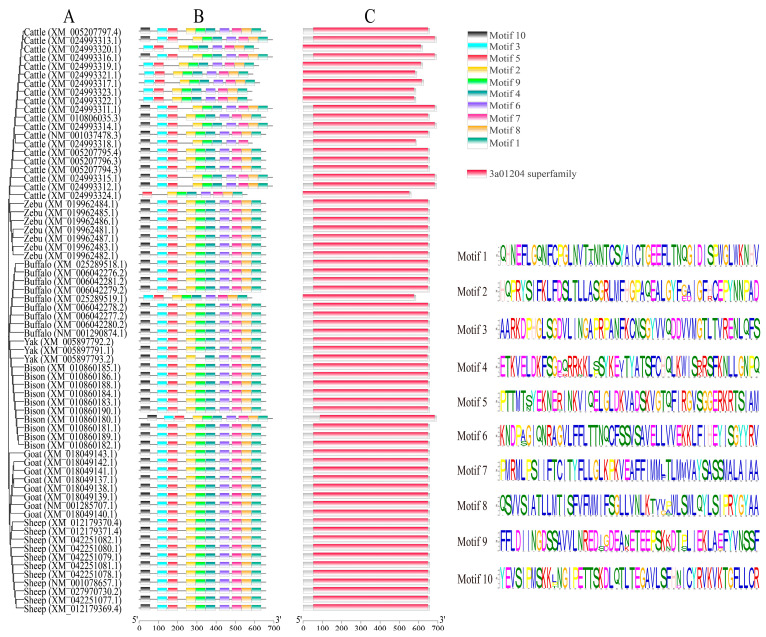
Phylogenetic relationships, motifs, and conserved domains of *ABCG2* in Bovidae species. (**A**) Phylogenetic tree; (**B**) motif pattern; (**C**) conserved domain. The conserved motifs and domains in *ABCG2* are marked with different color boxes.

**Figure 3 animals-13-03156-f003:**
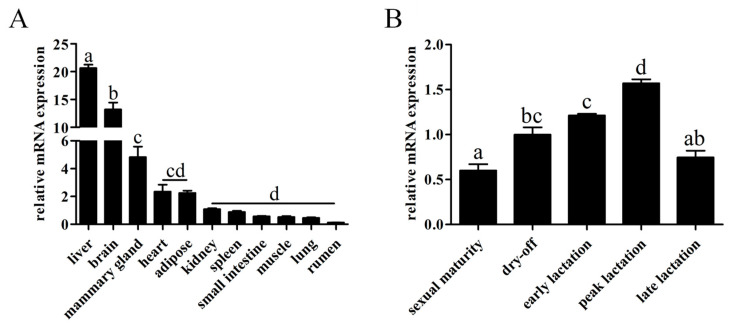
The mRNA expression profiles of buffalo *ABCG2*. (**A**) mRNA expression of *ABCG2* in various tissues in lactating buffalo. (**B**) Differential expression of *ABCG2* in different stages of mammary gland. The results are presented as mean ± SEM from *n* = 5 independent individuals; the different letters (a–d) represent significant differences (*p* < 0.05) in mRNA expression.

**Figure 4 animals-13-03156-f004:**
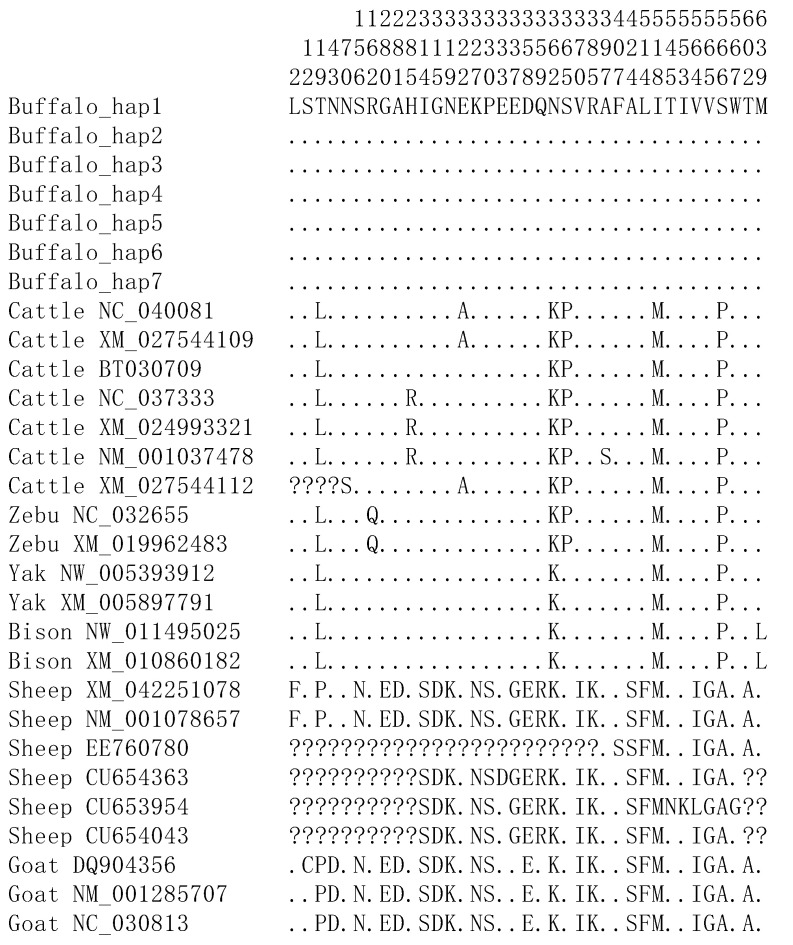
Differences in amino acid sequences corresponding to the *ABCG2* haplotypes in some species of Bovidae. Number represents the position of coding region. Dots (.) denote identity with Buffalo_hap1. Amino acid substitutions are denoted by different letters. Missing information is denoted by a question mark (?).

**Figure 5 animals-13-03156-f005:**
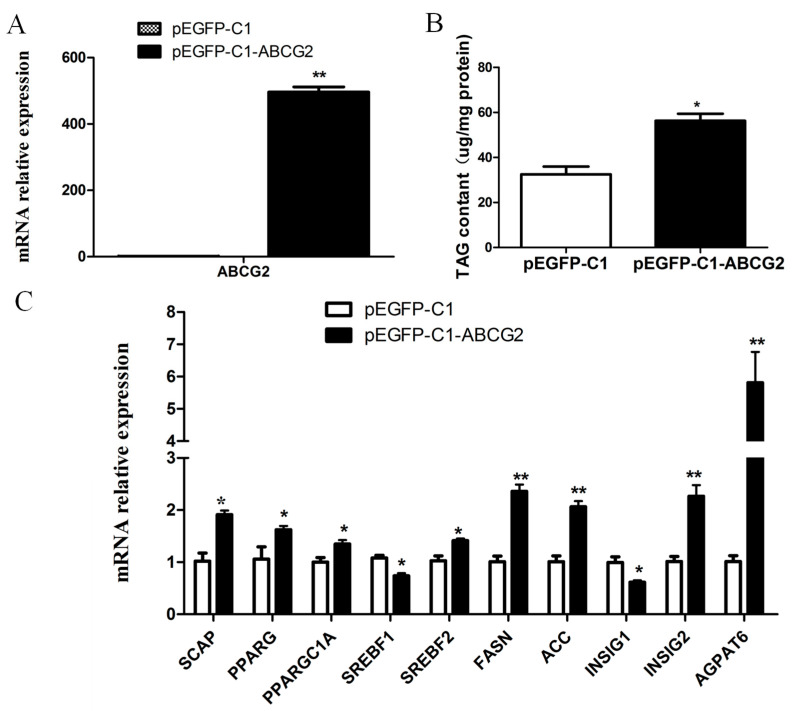
Effects of *ABCG2* overexpression on the genes related to milk fat synthesis and TAG content in BuMECs. (**A**) The expression of *ABCG2* in BuMECs after transfection of pEGFP-C1 or pEGFP-C1-ABCG2, respectively. (**B**) The change in TAG content in BuMECs after *ABCG2* overexpression. (**C**) The expression of genes related to milk fat synthesis after *ABCG2* overexpression. The results are presented as mean ± SEM from *n* = 3 independent cultures. * *p* < 0.05; ** *p* < 0.01.

**Figure 6 animals-13-03156-f006:**
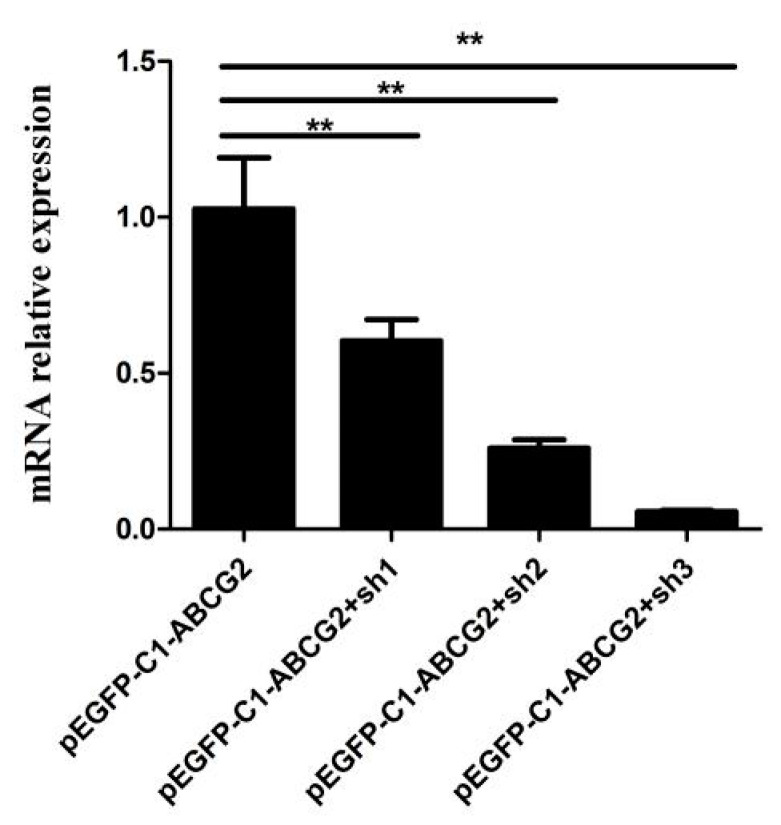
Interference efficiency of three shRNAs for *ABCG2* evaluated via qPCR. The results are presented as mean ± SEM from *n* = 3 independent cultures; ** *p* < 0.01.

**Figure 7 animals-13-03156-f007:**
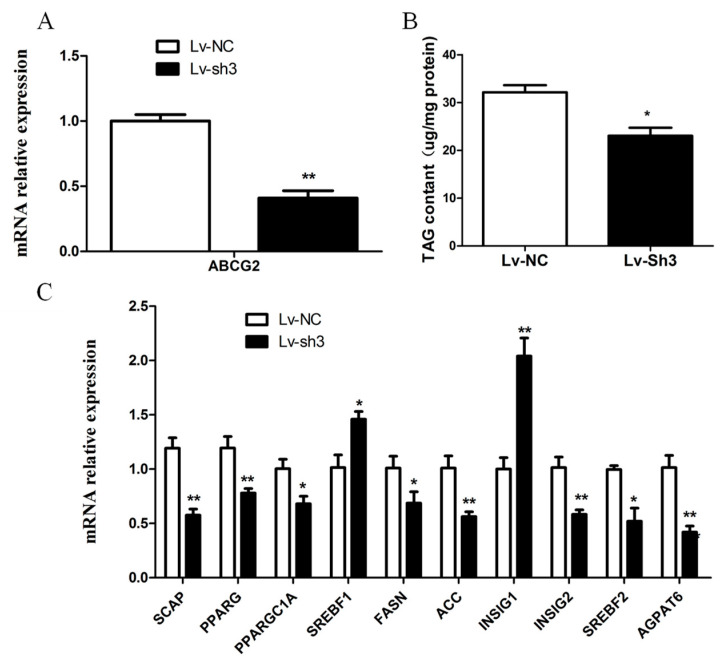
Effects of the *ABCG2* knockdown on the genes related to milk fat synthesis and TAG content in BuMECs. (**A**) The expression of *ABCG2* in BuMECs after being treated with Lv-sh3. (**B**) The change in TAG content in BuMECs after *ABCG2* knockdown. (**C**) The expression of genes related to milk fat synthesis in BuMECs after *ABCG2* knockdown. The results are presented as mean ± SEM from *n* = 3 independent cultures. * *p* < 0.05; ** *p* < 0.01.

**Table 1 animals-13-03156-t001:** Genetic information on the SNPs found in two types of buffaloes.

Population	SNP	Genotype Frequency			Allele Frequency		*p*-Value ^1^
		Genotype	Number	Frequency	Allele	Frequency	
River buffalo	c.393 C>T	CC	50	0.962	C	0.9808	0.9207
		CT	2	0.038	T	0.0192	
		TT	0	0.000			
	c.471 T>C	TT	49	0.942	T	0.9712	0.8618
		TC	3	0.058	C	0.0288	
		CC	0	0.000			
	c.720 C>T	CC	43	0.827	C	0.8942	0.0262
		CT	7	0.135	T	0.1058	
		TT	2	0.038			
	c.861 G>A	GG	47	0.904	G	0.9519	0.7458
		GA	5	0.096	A	0.0481	
		AA	0	0.000			
	c.1290 C>T	CC	51	0.981	C	0.9808	0.0000
		CT	0	0.000	T	0.0192	
		TT	1	0.019			
Swamp buffalo	c.393 C>T	CC	50	1.000	C	1.0000	–
		CT	0	0.000	T	0.0000	
		TT	0	0.000			
	c.471 T>C	TT	50	1.000	T	1.0000	–
		TC	0	0.000	C	0.0000	
		CC	0	0.000			
	c.720 C>T	CC	50	1.000	C	1.0000	–
		CT	0	0.000	T	0.0000	
		TT	0	0.000			
	c.861 G>A	GG	50	1.000	G	1.0000	–
		GA	0	0.000	A	0.0000	
		AA	0	0.000			
	c.1290 C>T	CC	50	1.000	C	1.0000	–
		CT	0	0.000	T	0.0000	
		TT	0	0.000			

^1^ *p*-value of Hardy–Weinberg equilibrium test.

## Data Availability

The data analyzed during the current study are available from the corresponding authors upon reasonable request.

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
