# Peer review of "Molecular Characteristics and Polymorphisms of Buffalo (Bubalus bubalis) ABCG2 Gene and Its Role in Milk Fat Synthesis"

_animals, 2023, doi:10.3390/ani13193156_

Round 1

Reviewer 1 Report

The author investigated the molecular characteristics and polymorphisms of buffalo ABCG2, which have a great importance of dairy buffalo industry. Nervelessness, the following concerns need to be addressed.

1. Line 17-18, please rewrite this sentence

2. ABCG2 appears in italics or not in different places in the article. Authors should make reasonable adjustments.

3. Line35. the TAG needs a full name.

4. Line 76, 80, and 87: the word (buffalo) needs plural. It is suitable for other position as well.

5. Line 92: why you select the cattle ABCG2 gene as the reference? Now, the buffalo ABCG2 gene is available.

6. Line 113-115, please list the species name and its GTF version.

7. Line 129, please cite it.

8. Line 133, please check the accession number. I cant find the gene sequence.

9. For the software used in the present study, the author should provide the version information

10. Line 150, I cant see the NC sequences in the Table S3.

11. In Figure 1. the author selected and compared the gene structure of different transcripts ABCG2 gene from different Bovidae species. In this study, a sequence of ABCG2 gene was cloned from a transcript, but what isoform is it part of? Figure 1 does n't clearly explain. More importantly, the design of the interference fragment of the buffalo ABCG2 gene, whether to use this sequence, is not clearly stated in the manuscript.

12. Line 273-279. the author should provide the sequencing results of ABCG2 variants. The readers cannot judge whether the ABCG2 variants provided by the author are accurate and reliable。

13. In Table 1. the author should provide the sample size for each genotype.

Not applicable

Reviewer 2 Report

Dear authors,

First of all, I want to congratulate you on your extensive work and on the very well written manuscript. 

I strongly believe that this work meets the criteria to be published, I have but a few suggestions:

- Aim, please rephrase it to be on past tense;

- Please insert a paragraph on each of the two buffalo breeds studied (e.g. census, history, production levels and if genetic selection programs are in place);

- I would advise you to mention the ethical statement in the Materials and Methods section as well, considering that there were animals sent to the slaughterhouse and also involved biopsy and blood sampling;

- Please highlight more in the manuscript your findings regarding the genotype frequency for the swamp buffalo (Table 1 and Lines 371-373), more explanations are needed, a simple statement does not suffice;

- Better highlight the practical implications from the breeding perspective of the two species.

Reviewer 3 Report

The manuscript describes a detailed analysis of the DNA sequence and expression of the ABCG2 gene in Buffalo. This gene potentially affects milk yield, therefore the presented work provides important information from the point of view of animal breeding. The work is written carefully and clearly. The methodology contains all the most important information. The results are clearly described and the discussion describes the most important information in the light of available literature data. I suggest some improvments:

line 35 - please explain TAG

Introduction: Please provide some information on the importance of Buffallo as a farm animal

Title: Please provide the latin name for Buffalo and use it through the manuscript

line 80-83were the sample collected from the same animals?

How were the animals (used for the SNP analysis) related?

line 200-203were tyhe distribution of qPCR data normal ? how was it checked?

line 213 What tissue was used for transcript analysis? How can you be sure that you identified all transcripts? Perhaps it would be better to say "all known transcripts"

Which parts of brains and kidneys were collected?

Figure 3 Please provide the number of animals for tissue analysis. Why the analysis of ABCG2 gene expression in the developmental stages of mammary gland was performed on cultures and nor directly on collected tissues?. Should not "dry off" in the end of the graph?

line 360-364 What could be the reason for the differences in tissue expression pattern in Buffalo and other bovidae species?

Round 2

Reviewer 1 Report

No comments

No comments